# The Role of the Wnt/β-Catenin Pathway in the Modulation of Doxorubicin-Induced Cytotoxicity in Cardiac H9c2 Cells by Sulforaphane and Quercetin

**DOI:** 10.3390/ijms26167858

**Published:** 2025-08-14

**Authors:** Viktória Líšková, Barbora Svetláková, Miroslav Barančík

**Affiliations:** Institute for Heart Research, Centre of Experimental Medicine, Slovak Academy of Sciences, Dúbravská cesta 9, 84104 Bratislava, Slovakia; viktoria.pecnikova@savba.sk (V.L.); barbora.svetlakova@savba.sk (B.S.)

**Keywords:** doxorubicin, cardiac cells, sulforaphane, quercetin, Wnt/β-catenin

## Abstract

This study investigates the role of sulforaphane (SFN) and quercetin (QCT) in alleviating the oxidative stress and modulation of cellular responses induced by doxorubicin (DOX) in rat cardiomyoblast cells H9c2. The potential mechanisms involving Wnt/β-catenin signaling and antioxidant response were determined. We found that SFN effectively mitigated DOX-induced cytotoxicity in H9c2 cells. These effects of SFN significantly exceeded the influence of QCT. Levels of superoxide dismutase isoforms 1 (SOD-1) and 2 (SOD-2) were upregulated following SFN and QCT pretreatment in cells exposed to effects of DOX. Additionally, β-catenin levels were increased following both SFN and QCT treatment, even in the presence of doxorubicin. Elevated β-catenin levels for QCT were associated with increased phosphorylation and inactivation of glycogen synthase kinase 3-β. The critical role of Wnt/β-catenin signaling in responses of H9c2 cells to effects of DOX was confirmed using Wnt/β-catenin inhibitor WIKI-4. This inhibitor increased the sensitivity of cells to DOX, and the decreased cellular viability after pretreatment with WIKI-4 was linked to SOD activities’ inhibition. Conclusively, sulforaphane and quercetin exert a protective effect against doxorubicin-induced cytotoxicity in H9c2 cells through the Wnt/β-catenin pathway as well as in association with modulation of enzymes related to the cellular antioxidant response.

## 1. Introduction

Doxorubicin (DOX) is a common anticancer drug that has been used in chemotherapy since the 1960s. This drug belongs to a group of anthracyclines and consists of a tetracycline ring complex attached to an amino sugar group that intercalates into DNA [1]. DOX has many clinical applications such as the treatment of breast, ovary, testicular, thyroid, and lung cancers and many types of sarcomas. Through intercalation into double-stranded DNA, it can disrupt DNA replication and RNA transcription, and it can also cause DNA damage by binding to the topoisomerase II enzyme and by induction of apoptosis. This drug, like all others, has side effects, of which the most studied is cardiotoxicity, which is a serious limitation of the doses of DOX in use [2,3]. The cardiotoxicity of DOX plays a role in ischemic heart disease, where it can lead to acute myocardial infarction [4], induce necrosis and apoptosis due to oxidative stress, and ultimately cause autophagic cell death [5,6].

The Wnt/β-catenin signaling pathway is a conserved cellular signaling system that is involved in embryonic development and the pathogenesis of human diseases. Under normal conditions, Wnt signaling is almost inactive, and β-catenin is phosphorylated and degraded by a destruction complex including Axin, adenomatous polyposis coli (APC), and glycogen synthase kinase-3β (GSK-3β) [7]. However, upon Wnt activation due to some pathological conditions, β-catenin translocates to the nucleus and then associates with TCF/LEF family transcription factors to activate target gene expression [8]. Under pathological conditions, several signaling pathways are activated, including Nrf-2 signaling. This pathway was linked to Wnt signaling in hepatocytes, where Wnt regulation of Nrf-2 was independent of Keap1 [9]. The authors of this study also observed that Wnt-3a disrupted Axin1 binding, GSK-3β phosphorylation, and β-TrCP ubiquitination of Nrf-2. Inoki et al. showed that, in cancer, Wnt signaling is involved in controlling the mTOR pathway’s activity through GSK-3β phosphorylation of TSC2 [10]. Initiation of this process is triggered by adenosine monophosphate-activated protein kinase (AMPK) in combination with GSK-3β, and AMPK phosphorylation stimulates the activity of TSC2 and the inhibition of mTOR [10]. Phosphorylation of GSK-3β leads to the inhibition of its kinase activity and subsequent accumulation of β-catenin. One of the ways to inhibit this kinase is via the PI3K/AKT signaling pathway, which phosphorylates GSK-3β at Ser9, and it was found that this pathway can be modulated by the Wnt ligand [11].

An essential factor in mechanisms that underlie the toxic effects of DOX is the oxidative stress induced by increased DOX-induced production of free radicals. Oxidative stress is one of the most common causes of pathological conditions in the body. It contributes to the development of many diseases such as inflammatory, cardiovascular, and neurodegenerative diseases or cancer. Various antioxidants were studied for their effects in lowering oxidative stress through modulation of the Nrf-2 signaling pathway [12,13,14]. One of those is sulforaphane (SFN), a natural antioxidant found in various vegetables, especially broccoli. Previous studies have demonstrated several positive effects of SFN, such as antidiabetic, anticancer, and cytoprotective effects [15,16]. Its involvement in the processes protecting the heart [17], kidneys [18], and brain [12] from ischemic damage has also been documented. Its mechanism of action is primarily associated with activation of the Nrf-2 (nuclear factor-erythroid factor 2-related factor 2) signaling pathway. This pathway is involved in the elimination of excess free radicals from the body through increased expression of antioxidant and detoxifying enzymes. SFN may also play an essential role in protecting cells from DOX action. It has been found that SFN can activate the Nrf-2-related signaling pathway, thereby affecting the expression of the genes regulated by Nrf-2 after DOX administration [19,20]. Signaling involving Nrf-2 controls the expression of several antioxidant and detoxifying enzymes, such as superoxide dismutase (SOD), heme oxygenase 1 (HO-1), and catalase (CAT) [21].

Another potential natural antioxidant is quercetin (QCT), which is found in several vegetables, such as onion [22]. QCT was described as a protector against inflammation, immunological dysregulation, and tumors [23,24]. It was found that QCT is probably a positive modulator of the Wnt/β-catenin signaling pathway. Predes et al. observed the increasing stability of β-catenin and the decreasing activity of GSK-3β [25]. QCT acts as a potent antioxidant that neutralizes reactive oxygen species (ROS) and thereby reduces oxidative stress. It activates endogenous antioxidant enzymes such as SOD, catalase, and glutathione peroxidase. In addition, it inhibits ROS-producing enzymes such as NADPH oxidase, thereby protecting cells from oxidative damage [26].

Recent studies have frequently discussed targeting Wnt signaling as a potential therapeutic strategy, including the use of small-molecule inhibitors, proteins, and peptide-based inhibitors. Wnt signaling can be blocked at multiple stages, for example by preventing the activity of Wnt ligands, inhibiting receptor phosphorylation, suppressing the function of Dishevelled (Dvl), stabilizing the β-catenin destruction complex, blocking the nuclear translocation of β-catenin, or interfering with the interaction between nuclear β-catenin and TCF transcription factors. Several classes of natural Wnt signaling antagonists are known to modulate downstream signaling, with the secreted Frizzled-related proteins (Sfrps), the Dickkopf (Dkk) family, and the Wnt inhibitory factor 1 (Wif1) being among the most extensively studied [27,28]. Inhibition of the Wnt/β-catenin signaling pathway with ICG-001 significantly reduced cardiac hypertrophy and fibrosis in mice with angiotensin II-induced hypertension. In the study of Zhao et al., the results suggest that the pharmacological blockade of this pathway represents a promising strategy for the prevention or treatment of hypertensive cardiomyopathy [29].

Our present study aimed to investigate the mechanisms underlying the effects of sulforaphane and quercetin on doxorubicin-induced toxicity in rat myoblast cell line H9c2. In connection with this, we investigated the impact of both SFN and QCT on proteins involved in the Wnt/β-catenin signaling pathway and antioxidant response related to redox signaling. The effects of WIKI-4, an inhibitor of the Wnt//β-catenin pathway, on effects induced by doxorubicin were also investigated.

## 2. Results

### 2.1. Effects of Sulforaphane and Quercetin on Modulation of Doxorubicin-Induced Cytotoxicity in H9c2 Cells

The cytotoxicity of the substances used in these experiments was determined in culture experiments performed on H9c2 cells. The cytotoxicity was evaluated by determining the IC_50_ value (‘inhibitory concentration’), which is a value indicating the concentration of the substance at which 50% of the cells survive. In the case of DOX, we found the IC_50_ value to be 0.8 µM; in the case of both SFN and QCT, we did not observe significant cytotoxic effects on H9c2 cells even at a concentration of 40 µM. Subsequently, we investigated how sulforaphane or quercetin affects DOX-induced cytotoxicity. We found that preculture of H9c2 cells with 10 µM SFN before DOX treatment significantly increased the viability of H9c2 cells in the presence of DOX when compared to the control DOX-treatment (Figure 1). Cells’ pretreatment with 20 µM QCT also increased the IC_50_ value to DOX, but this change was not significant unlike SFN.

### 2.2. Effects of Sulforaphane and Quercetin on Production of Cellular Reactive Oxygen Species

To further support the observed positive effects of SFN and QCT, we conducted an additional experiment focused on the determination of intracellular ROS levels. The data were analyzed using the Kruskal–Wallis test, which indicated statistically significant differences (*p* ≤ 0.05). We subsequently applied Dunn’s post hoc comparisons to assess individual group differences, resulting in *p* < 0.05 in six comparisons, four of which we consider biologically relevant. We found that both 1.25 and 2.5 μM DOX significantly increased the intracellular production of ROS (Figure 2). The effects of SFN and QCT on changes induced by DOX were not statistically significant. Still, we observed a consistent trend toward decreased ROS levels in samples that were preincubated with SFN and QCT compared to those treated with DOX alone. These findings suggest a potential antioxidant effect of both SFN and QCT.

### 2.3. Effects of Sulforaphane and Quercetin on Superoxide Dismutase and Catalase Activities

We analyzed how DOX, SFN, and QCT affect the activities of enzymes involved in cellular antioxidant defense—superoxide dismutase (SOD) and catalase (CAT). We also determined the influence of SFN or QCT pretreatment on the effects of DOX. The total enzyme activities were determined using commercially available SOD and catalase colorimetric assay kits. We found that DOX administration did not significantly influence the SOD activities compared to the control group. The SOD activities were also not modulated after SFN and or QCT treatment, alone or coadministered with DOX (Figure 3A,C).

As a result of DOX treatment, we found partial upregulation of CAT activities. However, pretreatment of cells with SFN before DOX application significantly decreased the CAT activities in the DOX + SFN group compared to the DOX group. Similar effects on DOX-induced changes in catalase activities were also found after pretreatment of cells with 20 µM QCT (Figure 3B,D). SFN alone markedly increased the catalase activities compared to control conditions (Figure 3B). In contrast, the effects of QCT were the opposite, and QCT alone decreased the catalase activities (Figure 3D).

### 2.4. Effects of Sulforaphane on the Modulation of Superoxide Dismutase Isoforms Protein Levels

The measurement of SOD activities provides information about the total SOD activities but does not distinguish between individual isoforms. To investigate the role of SOD isoforms in the effects of SFN and QCT, we determined the changes in protein levels of soluble cytoplasmic SOD-1 and mitochondrial SOD-2. We determined that SFN, despite not influencing total SOD activities, significantly modulated the protein levels of SOD-1 and SOD-2 isoforms (Figure 4A,B). Exposure of H9c2 cells to the effects of 10 µM SFN induced a significant increase in the protein levels of these enzymes. On the other hand, DOX did not influence the protein expression of SOD-1 and SOD-2, and pretreatment of cells with SFN moderately increased the protein levels of SOD-2 in the presence of DOX (Figure 4B).

### 2.5. Effects of Quercetin on the Modulation of Superoxide Dismutase Isoforms Protein Levels

Similarly to the effects of SFN, quercetin alone induced significant upregulation of both SOD-1 and SOD-2 protein levels (Figure 5). In contrast to the effects of sulforaphane, the increase in SOD isoforms protein levels was observed not only when QCT was administered alone but also in the conditions when cells were exposed to both QCT and DOX (Figure 5A,B).

### 2.6. Effects of Sulforaphane and Quercetin on Doxorubicin-Induced Modulation of Beta-Catenin Protein Levels

β-catenin represents an integral part of Wnt/β-catenin signaling, which plays a role in cellular responses to cellular stress. We found that pretreatment of H9c2 cells with both SFN (Figure 6A) and QCT (Figure 6B) induced upregulation of protein levels of β-catenin in DOX-treated cells. Sulforaphane or quercetin alone also induced significant upregulation of β-catenin protein levels.

### 2.7. Effects of Sulforaphane and Quercetin on Doxorubicin-Induced Modulation of GSK-3α/β Protein Levels and Specific Phosphorylation

GSK-3β is an enzyme that plays a role in regulating β-catenin levels. Phosphorylation of β-catenin by this enzyme is responsible for its increased proteolytic degradation. We determined that pretreatment of H9c2 cells with 10 µM SFN before application of 2.5 DOX µM did not significantly modulate the protein levels of GSK-3β. Treatment with SFN alone increased the levels of GSK-3β in comparison to the control conditions (Figure 7B). It was interesting to find that the effects of SFN on DOX-induced changes were different for GSK-3α. We discovered that the levels of GSK-3α were increased when the cells were pretreated with SFN. We also investigated changes in the specific phosphorylation of GSK-3β, but we did not observe significant changes in any experimental situation.

Treatment with QCT alone increased the protein levels of both GSK-3α and GSK-3β in comparison to the control conditions (Figure 7C,D). Moreover, QCT alone also increased the content of phosphorylated GSK-3β (Figure 7E). In contrast to the effects of SFN, quercetin did not influence the DOX-induced changes in GSK-3β protein levels. However, treatment of H9c2 cells with 20 µM QCT and 2.5 DOX µM increased the levels of phosphorylated GSK-3β in comparison to cells treated with DOX only (Figure 7E).

### 2.8. Influence of WIKI-4, a Wnt Pathway Inhibitor, on Doxorubicin-Induced Changes in H9c2 Cells

We found that the preculture of H9c2 cells with 5 or 10 µM WIKI-4 before DOX treatment significantly decreased the viability of H9c2 cells in the presence of DOX when compared to control DOX-treatment (Figure 8A). This increased sensitivity of cells to DOX (decreased viability of cells) was observed after pretreatment with 5 µM WIKI-4, which was associated with the significant inhibition of SOD activities (Figure 8B).

## 3. Discussion

The presented study shows that pretreatment with substances characterized by potential antioxidants properties possesses beneficial effects on DOX-induced cytotoxicity in myoblast H9c2 cells, as evident from their increased viability in the presence of DOX. Comparison of sulforaphane and quercetin in the mitigation of DOX-induced cytotoxicity showed that pretreatment with SFN significantly exceeded the effects of QCT.

Cytotoxicity induced through doxorubicin is primarily mediated by oxidative stress, mitochondrial dysfunction, and interference with topoisomerase IIβ in cardiomyocytes [30,31]. This often results in the increased generation of reactive oxygen species (ROS), lipid peroxidation, and activation of apoptotic and autophagic pathways. SFN and QCT are compounds with documented antioxidant effects [32,33], and in our presented study, we demonstrated their protective role against doxorubicin action in H9c2 cells.

To understand the molecular mechanisms underlying the modulatory role of sulforaphane and quercetin, we investigated changes in key regulatory proteins involved in β-catenin signaling and oxidative balance. An important finding was that both SFN and QCT showed the ability to modulate the expression of antioxidant enzymes such as SOD-1 and SOD-2, which aligns with previous studies reporting their capacity to upregulate endogenous antioxidant defense through Nrf-2 activation [34]. While increased protein expression of SOD-1 and SOD-2 was observed, the enzymatic activity of SOD remained unchanged. This discrepancy points to possible posttranslational regulation or reduced enzyme efficiency due to oxidative damage. Oxidative modifications, such as carbonylation or nitration, may lead to enzyme inactivation without reducing their abundance. Reduced enzymatic activity with increased expression may also be related to translation inhibition or degradation of the active protein [35]. Another possibility is impaired transport or protein misfolding, especially in the case of mitochondrial SOD-2 [36]. Doxorubicin can also chelate essential metal ions (Cu^2+^, Zn^2+^, Mn^2+^), which reduces the catalytic activity of SOD despite the presence of the protein [37].

Increased expression of SODs at the protein level probably represents a compensatory mechanism of the organism in response to increased oxidative stress. Still, it is not sufficient to change the overall enzymatic activity. In our experiments, where H9c2 cells were exposed to DOX and SFN or QCT, the Nrf-2 pathway is likely to be activated, which induces the transcription of antioxidant genes, including SOD [38]. However, despite the increased expression, persistent oxidative stress caused by doxorubicin may damage SOD proteins or alter their functionality. Although SFN and QCT may increase the expression of antioxidants, their effects may not be sufficient to overcome the toxic effects of DOX. Interference between DOX and sulfhydryl groups in the active site of the enzyme cannot be ruled out. The results point to the importance of monitoring not only the expression but also the functional activity of antioxidants when evaluating the effects of protective agents. Surprisingly, we observed completely different effects of SFN and QCT on catalase activity. When SFN was administered alone, there was a significant increase in catalase activity. These data are also supported by the results of studies that described a considerable role of SFN in modulating antioxidant defense, specifically through increasing catalase activity. In an experiment using H9c2 cells, SFN was shown to activate the transcription factor Nrf-2, which subsequently induced the expression of several antioxidant genes, including catalase, HO-1, and NAD(P)H dehydrogenase [quinone] 1. It has also been demonstrated that SFN can reduce doxorubicin-induced cellular oxidative stress via the Nrf-2/ARE signaling pathway, leading to increased stimulation of antioxidant mechanisms and cytoprotection against apoptosis [38,39,40]. Although SFN itself appears to be a positive modulator of the antioxidant response, in the case of DOX-induced effects, catalase activity was significantly decreased. We assume that SFN, as an exogenous antioxidant, can replace endogenous antioxidants in antioxidant defense against effects of DOX, since the enzyme activities were unchanged (SOD) or reduced (CAT). Still, cell viability was higher in the case of the combination of both DOX and SFN rather than in the case of DOX alone.

Unlike sulforaphane, quercetin reduced the activity of catalase in cells. Although it has been documented that QCT is also capable of modulating Nrf-2 [41], it also exhibits other complex effects that may depend on the dose and cellular environment. In addition to direct ROS scavenging, QCT may also act as a mild prooxidant and can chelate metal ions that are cofactors of heme enzymes such as catalase [42]. This chelation may lead to inhibition of catalase activity, even though enzyme expression may be increased. Overall, the differences in our observations may be due to the specific properties and mechanisms of action of sulforaphane and quercetin. SFN appears to be a potent inducer of the antioxidant response via Nrf-2. At the same time, the actions of QCT are more complex and may also involve direct chemical interactions leading to enzyme inhibition. These findings are consistent with the existing literature, which emphasizes the need to consider dose, cell type, and the environment when interpreting the antioxidant effects of natural polyphenols [43,44]. When measuring ROS levels, we determined that their amount significantly decreased after administration of both antioxidants in doxorubicin-induced cytotoxicity. This effect was associated with a decrease in catalase activity. Our results may indicate that the presence of antioxidants leads to an effective reduction in ROS levels, thereby reducing the need for catalase activation as an enzyme that breaks down hydrogen peroxide. Another possible explanation is that antioxidants directly neutralize ROS and other peroxide species before they are metabolized by catalase, thereby reducing its utilization and activity [42,45]. It is also possible that the activation of the antioxidant response (Nrf-2 signaling) leads to the preferential synthesis and subsequent activation of some antioxidant enzymes at the expense of others [46]. These findings point to the complexity of the antioxidant response and the need to consider the dynamics of individual defense mechanisms in combination therapy.

During our observation of changes in Wnt/ß-catenin signaling activation, we monitored changes in the protein expression of β-catenin, GSK-3α, GSK-3β, and GSK-3 phosphorylation. Our results showed that SFN and QCT administered separately led to an increase in GSK-3β expression. This finding may be attributed to the known role of GSK-3β as a key regulator of cell survival and apoptosis, when its expression is influenced by both oxidative stress and survival signaling pathways. Previous studies have shown that flavonoids, including quercetin, can affect GSK-3β either directly or indirectly through upstream regulators [47]. This is consistent with the key role of this enzyme in various signaling cascades, including Wnt, PI3K/Akt, and redox Nrf-2 signaling [48]. However, an interesting finding was that the combination of DOX with QCT or SFN did not result in changes in GSK-3β expression. This suggests that doxorubicin may activate pathways that attenuate or abrogate the effects of these phytocompounds on GSK-3β expression. DOX is known to induce extensive oxidative stress and apoptosis, which may affect the regulation of the expression of multiple kinases, including GSK-3β [49].

Furthermore, elevated β-catenin expression in cells treated with SFN and QCT supports the idea of canonical Wnt signaling activation. This pathway is generally silent in adult cardiomyocytes but becomes reactivated during stress, injury, or regeneration [50]. Although activation of Wnt/β-catenin signaling has been linked to both adaptive and maladaptive cardiac remodeling [51], our results suggest that modulation of this pathway by SFN and QCT might ensure protection under acute oxidative stress conditions. We also observed that the phosphorylated GSK-3β was significantly increased in QCT-treated cells, both alone and in combination with DOX. Phosphorylation of GSK-3β at Ser9 is a well-established mechanism of its inactivation, leading to accumulation of β-catenin and subsequent activation of Wnt target genes [52,53]. This suggests that QCT may promote β-catenin signaling through GSK-3β inhibition, possibly enhancing cell survival in the presence of DOX-induced toxicity. However, we observed that the combination of SFN or QCT with DOX resulted in increased β-catenin protein levels, although the GSK-3β protein levels and phosphorylation were unchanged under these conditions. This suggests that β-catenin may also be regulated by GSK-3 β-independent mechanisms, such as inhibition of other members of the degradation complex (e.g., Axin or APC), changes in active transcription or through DOX-induced oxidative stress, which itself may activate the Wnt/β-catenin pathway as a compensatory survival mechanism. GSK-3α expression was increased in our experimental conditions with QCT alone and with the combination of SFN and DOX. GSK-3α is a less studied isoform, but studies suggest its involvement in the regulation of cell proliferation and differentiation, and its inhibition may be related to protection against cardiac hypertrophy and remodeling [54]. Increased GSK-3α expression in the presence of both SFN and DOX may represent an adaptation mechanism of cells to stress conditions and promote cytoprotection in situations where GSK-3β phosphorylation is not activated. Overall, our observations support the hypothesis that flavonoids, such as QCT, and isothiocyanates, such as SFN, may modulate GSK-3 signaling in distinct ways, which may be important in the development of new strategies to protect cardiomyocytes from chemotherapeutic toxicity. The increased expression of β-catenin in the context of our findings therefore supports the hypothesis that SFN and QCT exert cytoprotective effects also through activation of the Wnt/β-catenin pathway. This activation may be synergistic with the effects of SFN and QCT on GSK-3 signaling. In this way, GSK-3 and β-catenin appear to be closely linked points in the protective signaling pathways regulated by SFN and QCT.

The critical role of Wnt/β-catenin signaling in responses of H9c2 cells to the effects of DOX was confirmed using Wnt/β-catenin inhibitor WIKI-4. This inhibitor increased the sensitivity of cells against DOX, and the decreased cellular viability after pretreatment with WIKI-4 was linked to the inhibition of SOD activities. This indicates that Wnt/β-catenin signaling may have a protective role in cardiomyocytes exposed to oxidative insults and that its inhibition could sensitize cells to oxidative damage and reduce cell survival. The effects of WIKI-4 support the hypothesis that an active Wnt/β-catenin pathway serves as a protective and compensatory mechanism in DOX-induced oxidative stress and apoptosis. The finding that WIKI-4 suppresses β-catenin activation and SOD is consistent with the results of a study showing that Dickkopf-1 (Dkk1), a Wnt inhibitor, exacerbates DOX-induced stimulation of apoptosis and mitochondrial damage in H9c2 cells and in vivo experiments [55]. On the other hand, the results of some studies suggest that inhibition of Wnt signaling may be beneficial—for example, short-term systemic inhibition of the Wnt/β-catenin pathway after myocardial infarction in mice improves tissue repair, reduces fibrosis, and promotes functional recovery [56]. Our observations of the loss of viability and decreased SOD activation upon Wnt inhibition correspond to the model scenario of the increased sensitivity of cells to oxidative stress upon DOX treatment. At the same time, it should be considered that long-term or excessive β-catenin activation may promote fibrotic processes and apoptosis in the heart. It has been shown that overexpression of β-catenin in H9c2 cells induced apoptosis and increased fibrosis markers [57]. This suggests that the role of Wnt/β-catenin signaling in responses of cardiac cells to stress conditions is ambiguous—its activation may be cytoprotective in acute stress situations such as DOX-toxic effects. At the same time, its excessive or chronic induction may induce undesirable effects associated with reduced cell survival.

Our findings demonstrate that sulforaphane (SFN) and quercetin (QCT), two natural compounds with well-documented antioxidant properties, exert cytoprotective effects against doxorubicin (DOX)-induced cytotoxicity in H9c2 myoblasts. Pretreatment with SFN significantly improved cell viability compared to QCT, highlighting its superior protective potential. Protective effects of both compounds were associated with the Wnt/β-catenin pathway and the modulation of enzymes related to cellular antioxidant response.

The observed findings indicate that GSK-3 and β-catenin are closely linked points in the cellular adaptation and survival signaling regulated by SFN and QCT. The crucial compensatory role of the β-catenin pathway under DOX-induced oxidative stress was also documented by the increased sensitivity to DOX upon its inhibition with WIKI-4. However, the dual nature of Wnt/β-catenin signaling in cardiomyocytes must be considered when considering therapeutic applications.

Both compounds modulated the expression of key antioxidant enzymes, although the enzymatic activities of these proteins did not always correspond with expression levels. This discrepancy underscores the importance of evaluating both the expression and functional activity of antioxidant enzymes in oxidative stress models.

Conclusively, the findings not only support the therapeutic potential of SFN and QCT in cardioprotection but also highlight the complexity of redox and survival pathways involved in their effects against cytotoxicity induced by doxorubicin. Further studies are warranted to delineate the long-term impact and clinical relevance of these phytochemicals in cardiotoxicity prevention.

## 4. Materials and Methods

### 4.1. Materials

D,L sulforaphane (SFN), quercetin (QCT), DCFDA, PMA, and Hoechst 33342 were purchased from Merck (Merck Life Science, Bratislava, Slovakia). WIKI4 (ab147053), a Wnt/beta catenin inhibitor, was obtained from Abcam (Cambridge, UK). Doxorubicin hydrochloride was obtained from Ebewe. Dimethyl sulfoxide (DMSO), Dulbecco’s modified high glucose Eagle’s medium (DMEM), fetal bovine serum (FBS), and sterile PBS were purchased from Merck (Merck Life Science, Bratislava, Slovakia). Glutamine–penicillin–streptomycin 100× and trypsin-EDTA 10× were purchased from Biosera (Kansas City, MO, USA). The colorimetric SOD assay kit (ab65354) and catalase assay kit (ab83464) were purchased from Abcam (Cambridge, UK).

Primary antibodies against GSK-3α/β (sc-7291), phospho-GSK-3β (sc-373800), and phospho-β-catenin (sc-57535) were purchased from Santa Cruz Biotechnology, Inc. (Dallas, TX, USA). Primary antibodies against β-catenin (#9562), SOD-1 (#2770), and SOD-2 (#13141) were purchased from Cell Signaling Technology (Danvers, MA, USA). Secondary peroxidase-labeled anti-rabbit or anti-mouse antibodies were purchased from Cell Signaling Technology (Danvers, MA, USA).

### 4.2. Cell Culture and Treatment

The H9c2 rat myoblast cell line (obtained from The European Collection of Authenticated Cell Cultures) was grown in DMEM supplemented with 10% FBS and 1% penicillin–streptomycin (Biosera, Kansas City, MO, USA). Cells were cultured a humidified atmosphere at 37 °C with 5% CO_2_. Passaged cells were washed with sterile PBS and PBS containing 0.25% trypsin, and 0.53 mm EDTA was used to release them from the bottom of culture flasks. After 5 min, we added DMEM high glucose complete medium to stop the trypsinization, and aliquots of cells were resuspended for further cultivation in complete culture medium.

### 4.3. Determination of Effects of Doxorubicin on Viability of H9c2 Cells

For these experiments, we added 100 µL of cell suspension and 100 µL of medium containing DOX to a 96-well plate. To monitor cytotoxicity, we used DOX in the concentration range of 0–5 µmol/L. After 22 h of culturing the cells in the presence of DOX, we determined the DOX cytotoxicity using the MTT assay. Experiments were performed independently three times in triplicate.

### 4.4. Determination of the Influence of Sulforaphane, Quercetin, and WIKI-4 on Doxorubicin-Induced Cytotoxicity

To investigate the effect of test substances (SFN, QCT, and WIKI-4) on the cytotoxicity of DOX, we precultured the cells for 2 h with the respective test substance. For cell pretreatment, we used 10 µmol/L sulforaphane, 20 µmol/L quercetin, and WIKI-4 at concentrations of 5 or 10 µmol/L. After pretreatment, we added DOX in the concentration range of 0–5 µmol/L. In control conditions, cells were precultured for 2 h before DOX treatment in the presence of 0.1% DMSO (vehicle for SFN, QCT, and WIKI-4). After a further 22 h of cell culture, we determined the cell viability using the MTT assay. We analyzed the cytotoxicity of DOX (IC_50_ value for DOX) in the absence and presence of SFN, QCT or WIKI-4. Experiments were performed independently at least three times in triplicate.

### 4.5. MTT Assay to Analyze Cell Viability

The impact of DOX, SFN, QCT, and WIKI-4 on the viability of H9c2 cells was determined using the MTT [3-(4,5-dimethylthiazol-2-yl)-2,5-diphenyl-tetrazolium bromide] assay on a 96-well plate. After 22 h of culturing the cells in the presence of DOX and/or test substances (SFN, QST, WIKI-4), 30 μL of 1 mg/mL MTT reagent in sterile PBS was added to the wells. After exposure for 2.5 h, the formed formazan crystals were dissolved in 100 µL of dimethyl sulfoxide. The absorbance of each well on the plate was detected at 570 nm using a microplate ELISA reader (Synergy H1 microplate reader, Biotek, Santa Clara, CA, USA).

We analyzed the cytotoxicity of DOX (IC_50_ value for DOX) in the absence and presence of SFN, QCT, or WIKI-4.

### 4.6. Samples Preparation

For biochemical analysis, the cells were exposed to the effects of SFN (final concentration of 10 μmol/L), QCT (final concentration of 20 μmol/L), or WIKI-4 (final concentration of 5 μmol/L) for 2 h. Then, we added DOX at the final concentration of 2.5 μmol/L. In control conditions, cells were precultured for 2 h before DOX treatment, in the presence of DMSO in the same concentration as each SFN, QCT, or WIKI-4. The cells were in a culture Petri dish, and after further 22 h, PBS containing 0.25% trypsin and 0.53 mm EDTA was used for 5 min to release the attached cells from the bottom of the culture dish. After 5 min of centrifugation at 1680× *g*, the sediment was resuspended in 300 μL of 50 mm Tris-HCl, pH 7.8, containing 0.1% Triton-X100. Subsequently, the obtained lysate was homogenized and centrifuged for 5 min at 2300× *g* at 4 °C. The supernatant obtained after this centrifugation was used for subsequent analyses. The protein concentrations in the samples were determined by the Bradford method.

### 4.7. Western Blot Analysis

For Western blot analysis, samples containing equivalent amounts of proteins in a single lane were separated by sodium dodecyl sulfate–polyacrylamide gel electrophoresis (SDS-PAGE). Before electrophoretic separation, isolated individual samples were mixed with reducing Laemmli sample buffer in a 2:1 ratio. These samples were heated for 5 min at 95 °C to denature the proteins. Proteins were separated by electrophoresis; the total amount of proteins loaded into the gel was 10 µg. After electrophoretic separation, the proteins were transferred to nitrocellulose membranes. After blocking nonspecific binding sites with 5% non-fat milk in Tris Buffered Saline (TBS), the membranes were incubated overnight at 4 °C with the appropriate specific primary antibody. The corresponding peroxidase-labeled anti-rabbit or anti-mouse immunoglobulins were used as secondary antibodies. Peroxidase reactions were detected by an enhanced chemiluminescence (ECL) system and quantified using Carestream software MI SE version 5.0.2.30 (Carestream Health, Woodbridge, CT, USA).

### 4.8. Determination of Superoxide Dismutase Activity

SOD activity in the samples was determined using a colorimetric superoxide dismutase activity assay kit (ab65354, Abcam, Cambridge, UK) according to the manufacturer’s protocol. The assay kit uses the water-soluble tetrazolium salt WST-1, which when reduced with a superoxide anion produces a formazan dye. The higher the SOD activity in the sample, the less formazan dye is formed. The production of formazan was determined by measuring the absorbance at 440 nm.

### 4.9. Determination of Catalase Activity

Catalase activity in the samples was determined using a colorimetric catalase activity assay kit (ab83464, Abcam, Cambridge, UK). This assay kit uses hydrogen peroxide (H_2_O_2_), which reacts with catalase to produce water and oxygen. In contrast, unconverted H_2_O_2_ reacts with the kit probe to form the product, the production of which was determined by measuring the absorbance at 570 nm. The determined catalase activity is reversely proportional to the signal.

### 4.10. Determination of Cellular ROS by DCFDA Assay

To evaluate the effects of sulforaphane (SFN) and quercetin (QCT) on doxorubicin (DOX)-induced cytotoxicity and reactive oxygen species (ROS) production, cells were pretreated with the test compounds in black 96-well plates. Cells were preincubated for 2 h with either 10 µM SFN or 20 µM QCT. Following this pretreatment, DOX was added at concentrations 1.25 and 2.5 µM along with the fluorescent ROS probe H2DCFDA at a final concentration of 50 µM. The cells were then incubated for an additional 22 h.

Subsequently, Hoechst 33342 was added at a concentration of 0.01 mg/mL, and the cells were incubated for 10 min to allow nuclear staining. Following incubation, the cells were washed with 200 µL phosphate-buffered saline (PBS) and treated with 200 µL of a DMSO:PBS solution (9:1, *v/v*) to facilitate probe solubilization.

Intracellular ROS levels were quantified by measuring the fluorescence at excitation/emission wavelengths of 495/525 nm for DCF and 361/461 nm for Hoechst 33342 using a microplate reader. Phorbol 12-myristate 13-acetate (PMA), a known ROS inducer, was used as a positive control. As a control for the solvent effects, cells were preincubated for 2 h with 0.1% DMSO—the vehicle used for both SFN and QCT—prior to DOX treatment.

### 4.11. Statistical Evaluation

The results are presented as the mean values ± standard error of the mean (SEM). For analysis of data related to intracellular ROS levels and cell viability, the Kruskal–Wallis test was used, which indicated statistically significant differences (*p* ≤ 0.05). We subsequently applied Dunn’s post hoc comparisons. For analysis of other data, we used one-factor ANOVA to detect statistically significant differences between groups. Differences between individual groups were determined using Tukey’s post hoc test. Differences were considered significant at *p* < 0.05.

## Figures and Tables

**Figure 1 ijms-26-07858-f001:**
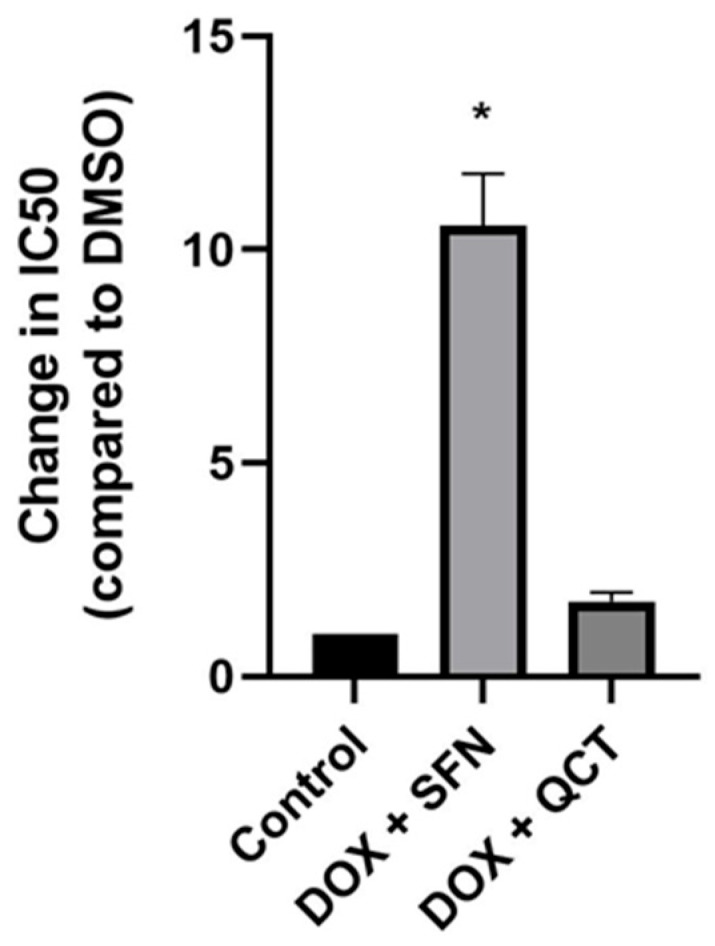
Effect of sulforaphane and quercetin on changes in cytotoxicity of DOX. Quantitative analysis of changes in the IC_50_ value for DOX after precultivation of H9c2 cells with 10 µM sulforaphane or 20 µM quercetin. Statistical significance was analyzed using the Kruskal–Wallis test with the subsequent Dunn’s post hoc comparisons. Each bar represents mean ± SEM. * *p* < 0.05 compared to control conditions (pretreatment with DMSO). Data were obtained from at least three independent experiments performed in triplicate.

**Figure 2 ijms-26-07858-f002:**
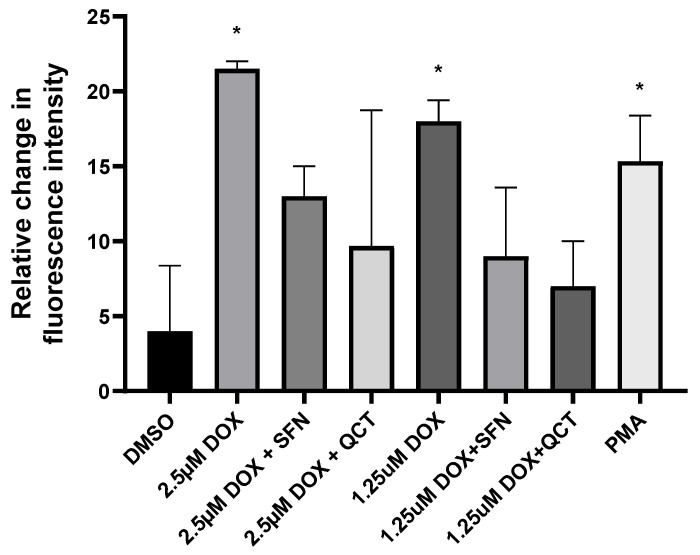
Effect of sulforaphane or quercetin on intracellular ROS production in H9c2 cells. Cells were pretreated with the test compounds in black 96-well plates. Cells were preincubated for 2 h with either 10 μM SFN or 20 μM QCT. Following this pretreatment, DOX was added at concentrations of 1.25 and 2.5 μM along with the fluorescent ROS probe H2DCFDA at a final concentration of 50 μM. The cells were then incubated for an additional 22 h. Subsequently, cells were incubated for 10 min with Hoechst 33342. Intracellular ROS levels were quantified by measuring fluorescence at excitation/emission wavelengths of 495/525 nm for DCF and 361/461 nm for Hoechst 33342 using a microplate reader. Each bar represents mean ± SEM, *n* = 3–4, * *p* < 0.05 compared to DMSO. DMSO—dimethyl sulfoxide; DOX—doxorubicin; SFN—10 μM sulforaphane; QCT—20 μM quercetin; PMA—phorbol 12-myristate 13-acetate (positive control, ROS inducer).

**Figure 3 ijms-26-07858-f003:**
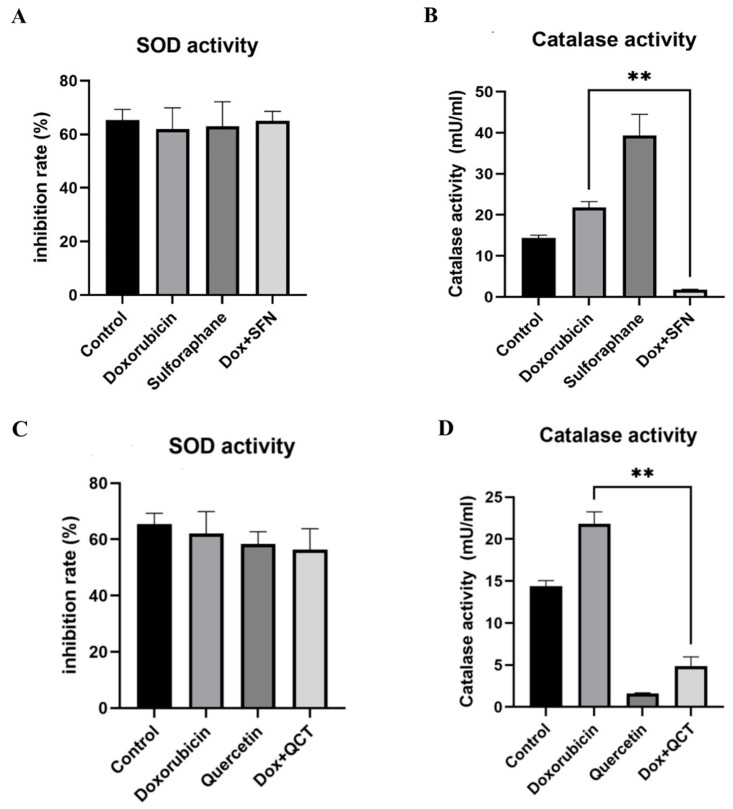
Effect of sulforaphane or quercetin on SOD and catalase activities in H9c2 cells. (**A**) SOD and (**B**) catalase activities in samples obtained after cell treatment with 10 µM SFN and/or 2.5 µM DOX were determined using colorimetric assay kits (Abcam). The data represent the percentage of inhibition of superoxide production by SOD activity. (**C**) Superoxide dismutase and (**D**) catalase activities in samples obtained after cell treatment with 20 µM QCT and/or 2.5 µM DOX were determined using colorimetric assay kit (Abcam). Each bar represents mean ± SEM, *n* = 4–6, ** *p* < 0.05 compared to doxorubicin group. For the statistical analysis, we used one-factor ANOVA. Control—0.1% DMSO; doxorubicin—2.5 µM; sulforaphane—10 µM; Dox + SFN—doxorubicin + sulforaphane pretreatment; Quercetin—20 µM; Dox + QCT—doxorubicin + quercetin pretreatment.

**Figure 4 ijms-26-07858-f004:**
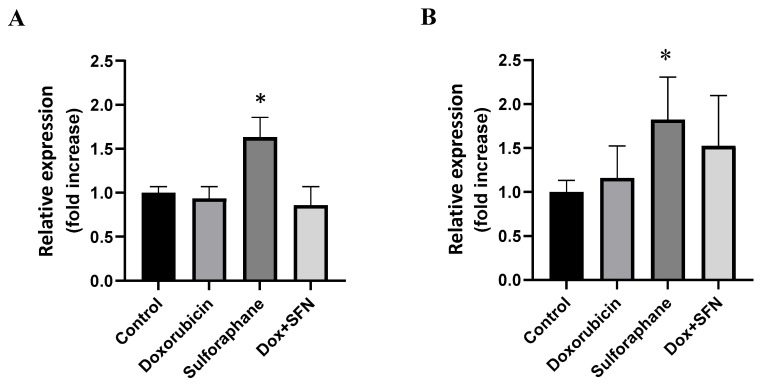
Effect of doxorubicin and/or sulforaphane on protein levels of SOD-1 and SOD-2 in rat H9c2 cells. (**A**) Quantitative analysis of changes in protein levels of SOD-1 and (**B**) SOD-2. Values are expressed as the relative intensity of the reaction compared to the control conditions. Statistical significance was analyzed by one-way ANOVA. Each bar represents mean ± SEM, *n* = 4–6, * *p* < 0.05 compared to control. Control—0.1% DMSO; doxorubicin—2.5 µM; sulforaphane—10 µM; Dox + SFN—doxorubicin + sulforaphane pretreatment.

**Figure 5 ijms-26-07858-f005:**
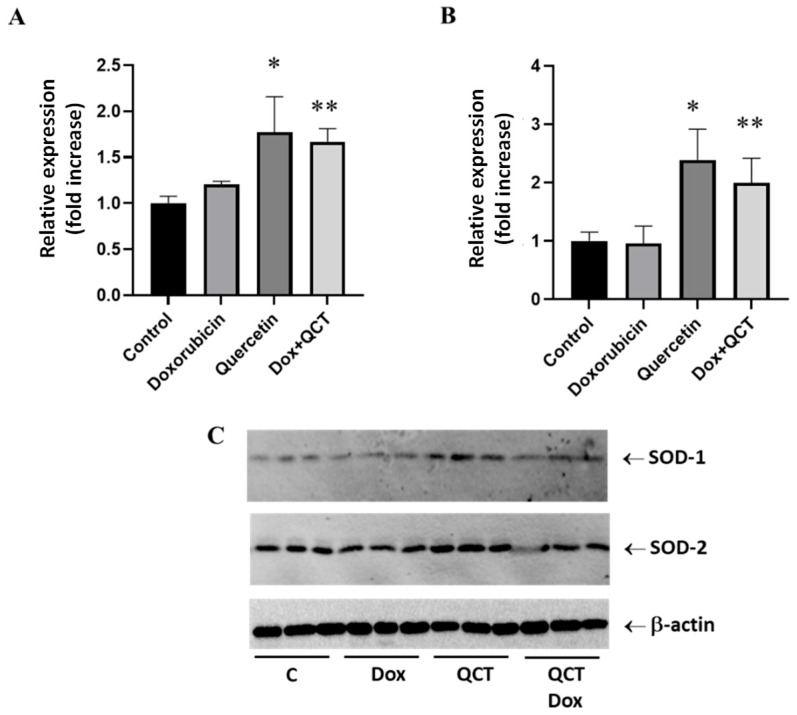
Effect of doxorubicin and/or quercetin on protein levels of SOD-1 and SOD-2 in rat H9c2 cells. (**A**) Quantitative analysis of changes in protein levels of SOD-1 and (**B**) SOD-2. Values are expressed as the relative intensity of the reaction compared to the control conditions. (**C**) Western blot records showing the protein levels of SOD-1 and SOD-2. Statistical significance was analyzed by one-way ANOVA. Each bar represents mean ± SEM, *n* = 4–6, * *p* < 0.05 compared to control, ** *p* < 0.05 compared to doxorubicin. Control—0.1% DMSO; doxorubicin—2.5 µM; quercetin—20 µM; Dox + QCT—doxorubicin + quercetin pretreatment.

**Figure 6 ijms-26-07858-f006:**
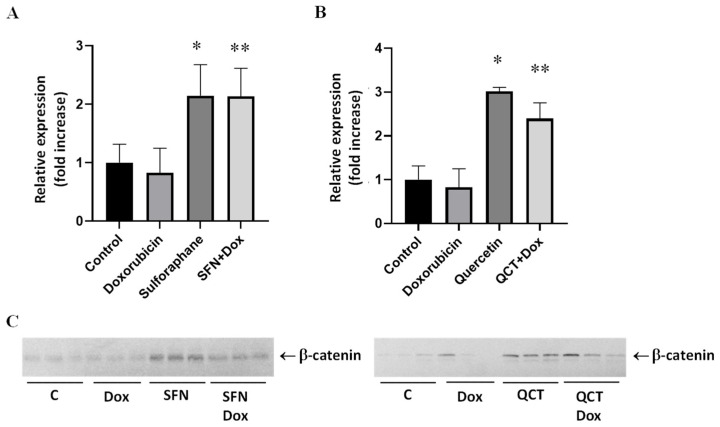
Effect of doxorubicin and/or sulforaphane on protein levels of beta-catenin in H9c2 cells. (**A**) Quantitative analysis of changes in β-catenin protein levels in samples obtained after cell treatment with 10 µM SFN and/or 2.5 µM DOX. (**B**) Quantitative analysis of changes in β-catenin protein levels in samples obtained after cell treatment with 20 µM QCT and/or 2.5 µM DOX. Statistical significance was analyzed by one-way ANOVA. Each bar represents mean ± SEM, *n* = 4–6, * *p* < 0.05 compared to control, ** *p* < 0.05 compared to doxorubicin. Control—0.1% DMSO; doxorubicin—2.5 µM; sulforaphane—10 µM; SFN+Dox—sulforaphane pretreatment + doxorubicin; quercetin—20 µM; QCT+DOX—quercetin pretreatment + doxorubicin. (**C**) Western blot records showing the protein levels of β-catenin. C—control; Dox—doxorubicin; SFN—sulforaphane; QCT—quercetin.

**Figure 7 ijms-26-07858-f007:**
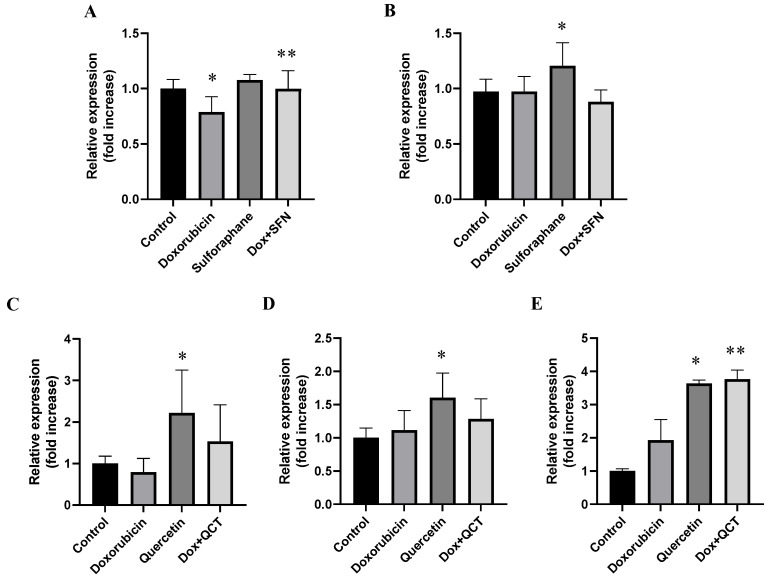
Effect of doxorubicin and/or sulforaphane on protein levels of glycogen synthase 3 alpha (GSK-3α) and glycogen synthase kinase 3 beta (GSK-3β) in H9c2 cells. Quantitative analysis of changes in (**A**) GSK-3α protein levels and (**B**) GSK-3β protein levels in samples obtained after cell treatment with 10 µM SFN and/or 2.5 µM DOX. Quantitative analysis of changes in (**C**) GSK-3α and (**D**) GSK-3β protein levels in samples obtained after cell treatment with 20 µM QCT and/or 2.5 µM DOX. (**E**) Changes in specific phosphorylation of GSK-3β. Statistical significance was analyzed by one-way ANOVA. Each bar represents mean ± SEM, *n* = 4–6, * *p* < 0.05 compared to control, ** *p* < 0.05 compared to doxorubicin group. Control—0.1% DMSO; doxorubicin—2.5 µM; sulforaphane—10 µM; Dox + SFN—doxorubicin + sulforaphane pretreatment; quercetin—20 µM; Dox + QCT—doxorubicin + quercetin pretreatment.

**Figure 8 ijms-26-07858-f008:**
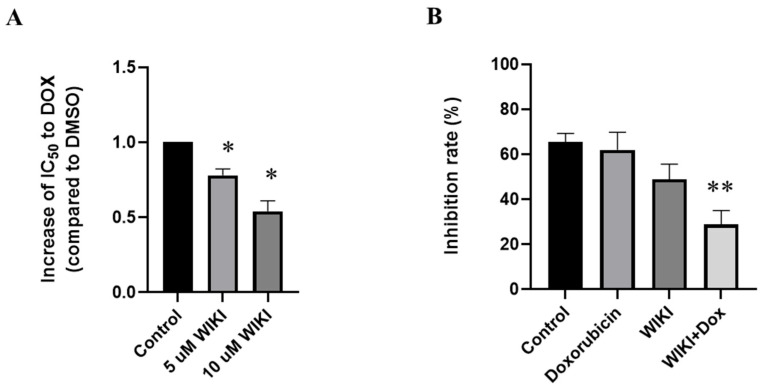
Effect of WIKI-4 on changes in cytotoxicity of DOX. (**A**) Quantitative analysis of changes in the IC_50_ value for DOX after precultivation of H9c2 cells with 5 or 10 µM WIKI-4. (**B**) Effect of WIKI-4 on SOD activities in H9c2 cells. SOD activities in samples obtained after cell treatment with 5 µM WIKI-4 and 2.5 µM DOX were determined using a colorimetric assay kit (Abcam). The data represent the percentage of inhibition of superoxide production by SOD activity. Each bar represents mean ± SEM, *n* = 4–6, * *p* < 0.05 compared to control group, ** *p* < 0.05 compared to doxorubicin group. Control—0.1% DMSO; doxorubicin—2.5 µM; WIKI—5 µM; WIKI+Dox—WIKI pretreatment + doxorubicin.

## Data Availability

The raw data supporting the conclusions of this article will be made available by the authors on request.

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
