# Peer review of "The Role of the Wnt/β-Catenin Pathway in the Modulation of Doxorubicin-Induced Cytotoxicity in Cardiac H9c2 Cells by Sulforaphane and Quercetin"

_ijms, 2025, doi:10.3390/ijms26167858_

Round 1
Reviewer 1 Report
Comments and Suggestions for Authors
The presented manuscript is devoted to the study of approaches to reducing the toxicity of doxorubicin. After reviewing the work, I had two main comments:
- The study lacks a comprehensive analysis of the data obtained. To increase the reliability of the conclusions, it is recommended to expand the methodological base by including additional experimental approaches. In particular, it is advisable to supplement the analysis of the expression of the studied markers not only at the protein level (using Western blotting), but also at the transcriptional level. This will provide a more complete picture of the molecular mechanisms of the processes under study.
- Insufficient scientific novelty and predictability of results. The data obtained demonstrate the expected effects, which reduces the originality of the study. To enhance the significance of the work, it is recommended to deepen the analysis, for example, to study additional aspects of the mechanism of action of the proposed approaches or to conduct a comparative analysis with alternative strategies for reducing toxicity.
I also have a number of short comments that could improve the work:
- The results sections (especially 2.3 and 2.4) require a brief introduction explaining the logic of the experimental design. A good example is section 2.6, where the structure of the presentation is clearer.
- In Figure 1, at a concentration of 20 μM, quercetin QCT does not show a significant effect on cell viability. For a better interpretation of the results, it is recommended to present the raw data on viability after treatment: with DOX alone, and with combinations of DOX and QCT.
- It is unclear what distribution the data presented in Figure 1 have. Justification for the use of analysis of variance (ANOVA) or an indication of the use of alternative nonparametric methods (for example, the Kruskal-Wallis test) in the case of a non-normal distribution is required.
- The legend to Figure 2 does not indicate the statistical analysis method used. This information should be added to ensure reproducibility of the results.
- The Western blot image quality in Figure 5 is insufficient for definitive conclusions. We recommend providing clearer original data or repeating the experiment.
- The presentation of results as Fold-increase in Figure 5 may mask significant variability between replicates. Considerations include: presenting the original data, checking the data for ANOVA, or using nonparametric analogs.
- The conclusions in section 2.7 are formulated correctly, but their discussion should be expanded, as this is of interest.
- To strengthen the evidence, it is necessary to directly demonstrate a reduction in reactive oxygen species after a combination of drugs, showing not only the "beneficial effect" but also its consequences.
Author Response
Response to reviewers’ comments
Reviewer 1:
Main comments:
Comment: The study lacks a comprehensive analysis of the data obtained. To increase the reliability of the conclusions, it is recommended to expand the methodological base by including additional experimental approaches. In particular, it is advisable to supplement the analysis of the expression of the studied markers not only at the protein level (using Western blotting), but also at the transcriptional level. This will provide a more complete picture of the molecular mechanisms of the processes under study.
RESPONSE: We thank reviewer for valuable comments. We fully agree that combining expression analysis at both the protein and transcriptional levels would provide a more comprehensive view of the molecular mechanisms underlying the effects of sulforaphane and quercetin. However, in this study, we focused only on changes at the protein level, as these reflect the functional state of cells and the direct biological effect of the substances under investigation. In the case of signaling proteins (GSK-3β, β-catenin), rapid post-translational modifications are often more important than gene expression itself. Another reason was that our study included a functional analysis of enzymatic activity (SOD, catalase), which naturally relates to the presence and amount of the enzymes in their active form, not to mRNA levels. Nevertheless, we consider the addition of transcription analysis to be beneficial and plan to include it in an extended set of experiments in the future.
Comment: Insufficient scientific novelty and predictability of results. The data obtained demonstrate the expected effects, which reduces the originality of the study. To enhance the significance of the work, it is recommended to deepen the analysis, for example, to study additional aspects of the mechanism of action of the proposed approaches or to conduct a comparative analysis with alternative strategies for reducing toxicity.
RESPONSE: We thank reviewer for constructive comment. We appreciate the reviewer’s critical evaluation of our manuscript and the comment regarding the perceived limited originality and predictability of the results.
We would like to point out that although the protective effects of antioxidants such as SFN and QCT have been described in the literature, our main ambition was to investigate their impact in the context of the Wnt/β-catenin signaling pathway in cardiomyoblasts, which is a poorly explored mechanism, especially in connection with DOX. Our study provides a new insight into the regulatory link between oxidative stress, Wnt pathway activation, and the expression of antioxidant enzymes (SOD isoforms, catalase). We see the originality mainly in the finding that the protective effects of SFN and QCT are associated with β-catenin activation even in the absence of changes in GSK-3β, suggesting GSK-3-independent mechanisms of Wnt pathway regulation – an aspect that has not been sufficiently described in this context. Another new insight is that outwardly similar antioxidants act through different mechanisms, which may have important implications for their therapeutic use and combination with other drugs. However, we agree that a better analysis of other aspects of the mechanism of action and the extension of our measurements would increase the significance of the work. Therefore, we plan to analyze the effects of SFN and QCT using other known modulators of the Wnt pathway and redox signaling, analyze the gene expression of target genes regulated by Wnt/Nrf2, and to investigate other intracellular signaling pathways with potential involvement in the mechanism of DOX cytotoxicity and the effects of SFN and QCT.
In summary, key novel contributions of our study include:
- Modulation of DOX cytotoxicity by SFN and QCT via Wnt signaling: While both sulforaphane (SFN) and quercetin (QCT) are known for their antioxidant properties, our data show that they increase the IC50 of doxorubicin (DOX) in H9c2 cardiomyoblasts in correlation with changes in the Wnt/β-catenin pathway. This suggests a protective, modulatory role that has not been mechanistically connected in this way before.
- Selective modulation of antioxidant enzyme expression vs. activity: We observed that SFN and QCT increase protein levels of SOD1 and SOD2, yet total SOD enzymatic activity remains unchanged. Moreover, catalase activity was reduced in the presence of DOX combined with SFN or QCT. This uncoupling of transcriptional regulation from enzymatic activity highlights complex redox regulation beyond classical antioxidant effects and adds mechanistic depth.
- Inhibition by WIKI4 reveals a functional role of Wnt signaling in sensitivity of H9c2 cells to DOX. Inhibition of this pathway reduces cytotoxicity of DOX and decreases SOD activity. This functional experiment supports the involvement of Wnt signaling in cytoprotection and antioxidant response.
Short comments:
Comment: The results sections (especially 2.3 and 2.4) require a brief introduction explaining the logic of the experimental design. A good example is section 2.6, where the structure of the presentation is clearer.
RESPONSE: We realized corresponding changes in revised manuscript. We included a brief introduction explaining the logic behind the experiment to the section 2.3 – lines 193-196.
Comment: In Figure 1, at a concentration of 20 μM, quercetin QCT does not show a significant effect on cell viability. For a better interpretation of the results, it is recommended to present the raw data on viability after treatment: with DOX alone, and with combinations of DOX and QCT.
RESPONSE: Re-evaluation of the data using the Kruskal-Wallis test revealed not statistically significant differences for QCT. Based on this analysis, we decided to present the results in their original form.
Comment: It is unclear what distribution the data presented in Figure 1 have. Justification for the use of analysis of variance (ANOVA) or an indication of the use of alternative nonparametric methods (for example, the Kruskal-Wallis test) in the case of a non-normal distribution is required.
Following your valuable comments regarding the choice of statistical tests used in the analysis of cell viability data, we revisited and reanalyzed the dataset using a more appropriate non-parametric approach. Specifically, we re-evaluated the data using the Kruskal-Wallis test, which still revealed statistically significant differences. However, not all differences previously indicated in the graphs were confirmed by the subsequent Dunn’s post hoc comparisons. Nevertheless, we would like to emphasize the consistent trend observed across both experiments, with p < 0.05 supporting the presence of significant changes.
Comment: The legend to Figure 2 does not indicate the statistical analysis method used. This information should be added to ensure reproducibility of the results.
RESPONSE: We have accepted the proposed comment. Statistical analysis method has been added to the legend to previous Figure 2 (in revised manuscript Figure 3).
Comment: The Western blot image quality in Figure 5 is insufficient for definitive conclusions. We recommend providing clearer original data or repeating the experiment.
RESPONSE: We have accepted the proposed suggestion and we have done required change in Figure 5 (in revised manuscript Figure 6). For sulforaphane we used Western blot record from previous experiments with clearer data, for QCT we repeated experiment.
Comment: The presentation of results as Fold-increase in Figure 5 may mask significant variability between replicates. Considerations include: presenting the original data, checking the data for ANOVA, or using nonparametric analogs.
The reason for using the presentation of results as a fold-increase over the control was the fact that samples were always compared with control within individual membranes. On each membrane were used 2-3 control samples and the samples were compared to mean value of these control sample. The protein expression in the control was taken as 1 and that of the other groups is shown relative to the mean of control group.
Comment: The conclusions in section 2.7 are formulated correctly, but their discussion should be expanded, as this is of interest.
RESPONSE: Based on your recommendation, we have expanded the section of the discussion concerning the inhibition of Wnt/b-catenin signaling (lines 417-435).
Comment: To strengthen the evidence, it is necessary to directly demonstrate a reduction in reactive oxygen species after a combination of drugs, showing not only the "beneficial effect" but also its consequences.
RESPONSE: We have accepted the proposed suggestion. We have done additional experiments related to the measurement of reactive oxygen species production after a combination of drugs. We included obtained data into revised manuscript. These data are presented in new Figure 2.
Reviewer 2 Report
Comments and Suggestions for Authors
The manuscript by Viktória Líšková et al investigates the protective effects of sulforaphane (SFN) and quercetin (QCT) against doxorubicin-induced cytotoxicity in H9c2 cardiomyoblasts, focusing on Wnt/β-catenin signaling and antioxidant mechanisms. While the topic is of considerable interest and relevance to cardio-oncology, this reviewer believes that several methodological and interpretative shortcomings limit the current impact and reliability of the findings.
Notably, there are concerns regarding the statistical analyses. ANOVA appears to have been used in contexts where the assumption of homogeneity of variances is not met—particularly when variability in the control group is minimal. Moreover, the interpretation of key findings remains largely descriptive and fails to fully explore the underlying mechanisms, such as the paradoxical reduction in catalase activity or the lack of change in SOD enzymatic function despite protein upregulation.
Unfortunately, the Western blot data are presented only as cropped bands, and some bands appear unclear or inconsistently displayed. For the sake of transparency and reproducibility, the full uncropped blots should be provided. Overall, the manuscript would benefit from major revisions to improve statistical rigor, enhance mechanistic interpretation, and ensure data transparency.
Major points
- Discussion Depth and Mechanistic Clarification
The current Discussion section lacks sufficient mechanistic interpretation of key findings. For example, although the discrepancy between increased SOD protein expression and unchanged enzymatic activity is acknowledged, potential explanations—such as post-translational modifications, oxidative inactivation, or protein misfolding—are not adequately addressed. Similarly, the observed decrease in catalase activity following antioxidant treatment appears paradoxical and should be discussed in greater depth, particularly in the context of the expected protective roles of SFN and QCT.
While the role of β-catenin and GSK-3β signaling is appropriately highlighted, the discussion would benefit from distinguishing between the acute protective and chronic maladaptive effects of Wnt/β-catenin activation in cardiomyocytes. In addition, consideration of upstream regulators (e.g., PI3K/Akt) and downstream effectors (e.g., FOXO, mTOR) would provide a more comprehensive understanding of the signaling context.
Finally, referencing prior studies with similar findings would enhance the interpretive framework and reinforce the biological relevance of the results.
- Statistical Methods and Validity
The statistical methodology requires further clarification and possible revision. For example, in Figures 1 and 7A, the control groups appear to exhibit minimal variability, yet ANOVA was applied. This raises concerns regarding the validity of the analysis, as ANOVA assumes homogeneity of variances. The authors should clarify whether the assumptions of normality and equal variances were tested prior to conducting parametric tests.
- Western Blot Presentation and Transparency
The Western blot data are presented only as cropped images, and some bands appear unclear—particularly those shown in Fig. 5. For the sake of transparency and reproducibility, the authors must provide the full-length, uncropped images of all Western blots used in the manuscript. In Fig. 5 specifically, the bands are difficult to interpret, and no loading control such as β-actin is shown, which raises concerns about the reliability of the quantitative data.
Minor Point
Formatting Consistency – Greek Letters
There are several instances throughout the manuscript (e.g., around line 200) where Greek letters such as β and γ appear to be incorrectly formatted or have been unintentionally converted to standard characters or symbols. The authors are advised to carefully review the entire text to ensure proper formatting of all Greek letters and special characters, particularly in protein names and signaling pathway components.
Author Response
Response to reviewers’ comments
Reviewer 2:
Comment 1: Notably, there are concerns regarding the statistical analyses. ANOVA appears to have been used in contexts where the assumption of homogeneity of variances is not met—particularly when variability in the control group is minimal.
RESPONSE: Following your valuable comments regarding the choice of statistical tests we have done revision of statistical methodology. By the analysis of the cell viability data and the data related to intracellular ROS levels we used a more appropriate non-parametric approach. We reevaluated these data using the Kruskal-Wallis test with subsequent application of Dunn’s post hoc comparisons For analysis of other data we used one-factor ANOVA.
Comment 2: Moreover, the interpretation of key findings remains largely descriptive and fails to fully explore the underlying mechanisms, such as the paradoxical reduction in catalase activity or the lack of change in SOD enzymatic function despite protein upregulation.
RESPONSE: We thank reviewer for constructive comment. In modified Discussion of revised manuscript we explained the possible mechanisms playing a role in observed effects of test substances on superoxide dismutase and catalase (lines 304-310 and 314-361). The changes are highlighted by red color.
Comment 3: Unfortunately, the Western blot data are presented only as cropped bands, and some bands appear unclear or inconsistently displayed. For the sake of transparency and reproducibility, the full uncropped blots should be provided. Overall, the manuscript would benefit from major revisions to improve statistical rigor, enhance mechanistic interpretation, and ensure data transparency.
RESPONSE: Full original blots are provided in supplementary materials. In revised manuscript we have done several modifications to improve statistical rigor and to enhance mechanistic interpretation (see also responses to major points).
Major points:
Comment 4: The current Discussion section lacks sufficient mechanistic interpretation of key findings. For example, although the discrepancy between increased SOD protein expression and unchanged enzymatic activity is acknowledged, potential explanations—such as post-translational modifications, oxidative inactivation, or protein misfolding—are not adequately addressed. Similarly, the observed decrease in catalase activity following antioxidant treatment appears paradoxical and should be discussed in greater depth, particularly in the context of the expected protective roles of SFN and QCT.
RESPONSE: We thank reviewer for constructive comment. In modified Discussion of revised manuscript we explained the possible mechanisms playing a role in observed effects of test substances on superoxide dismutase and catalase (lines 304-310 and 314-361). The changes are highlighted by red color.
Comment 5: While the role of β-catenin and GSK-3β signaling is appropriately highlighted, the discussion would benefit from distinguishing between the acute protective and chronic maladaptive effects of Wnt/β-catenin activation in cardiomyocytes.
RESPONSE: We thank reviewer for constructive comment. In revised manuscript we discussed the different role of Wnt/β-catenin activation in acute and chronic cellular responses (lines 423-435). The changes are highlighted by red color.
Comment 6: In addition, consideration of upstream regulators (e.g., PI3K/Akt) and downstream effectors (e.g., FOXO, mTOR) would provide a more comprehensive understanding of the signaling context.
RESPONSE: Thank you for your valuable comment. We tried to determine the effects of DOX, SFN, and QCT on changes in activation (specific phosphorylation) of Akt kinase. However, when we used antibodies specific to Akt kinase phosphorylated at Ser473, we did not observe any signal. Meanwhile, this problem did not occur in the analyses of samples from HEK293 cells or the left ventricular tissue of the rat heart.
In future experiments we consider to investigate more precisely individual components of signaling.
Comment 7: Finally, referencing prior studies with similar findings would enhance the interpretive framework and reinforce the biological relevance of the results.
RESPONSE: In discussion of manuscript are included references to several studies directly related to our findings and supporting these findings, such as references 38, 39, 40, and 55.
Comment 8: The statistical methodology requires further clarification and possible revision. For example, in Figures 1 and 7A, the control groups appear to exhibit minimal variability, yet ANOVA was applied. This raises concerns regarding the validity of the analysis, as ANOVA assumes homogeneity of variances. The authors should clarify whether the assumptions of normality and equal variances were tested prior to conducting parametric tests.
RESPONSE: Based on you recommendation we have done revision of statistical methodology. By analysis of cell viability data and the data related to intracellular ROS levels we used a more appropriate non-parametric approach. We evaluated these data using the Kruskal-Wallis test with subsequent application of Dunn’s post hoc comparisons For analysis of other data we used one-factor ANOVA.
Comment 9: The Western blot data are presented only as cropped images, and some bands appear unclear—particularly those shown in Fig. 5. For the sake of transparency and reproducibility, the authors must provide the full-length, uncropped images of all Western blots used in the manuscript. In Fig. 5 specifically, the bands are difficult to interpret, and no loading control such as β-actin is shown, which raises concerns about the reliability of the quantitative data.
RESPONSE: The protein loading was always controlled by Ponceau staining of nitrocellulose membranes after transfer. We did not observe changes in amount of proteins loaded per individual lanes. We have done also Western blot analysis with antibody against β-actin. Also these blots confirmed that there were not differences in protein loading.
We have done changes in Figure 5 (in revised manuscript Figure 6). We present Western blot data with more clear bands. For sulforaphane we used new Western blot record from previous experiments with clearer data, for QCT we repeated experiment. Full original blots are provided in supplementary materials.
Minor point:
Comment: There are several instances throughout the manuscript (e.g., around line 200) where Greek letters such as β and γ appear to be incorrectly formatted or have been unintentionally converted to standard characters or symbols. The authors are advised to carefully review the entire text to ensure proper formatting of all Greek letters and special characters, particularly in protein names and signaling pathway components.
RESPONSE: Thank you for the comment. We checked the manuscript and formatted correctly Greek letters and special characters in text.
Round 2
Reviewer 2 Report
Comments and Suggestions for Authors
This manuscript has been appropriately revised, and there are no remaining concerns form this reviewer.
Author Response
We thank reviewer for many suggestive ideas and comments that helped to improve the quality of our study.